# Performance of Urine Reagent Test Strips in Detecting *Schistosoma haematobium* Infection in Individual and Pooled Urine Samples

**DOI:** 10.3390/microorganisms13030510

**Published:** 2025-02-26

**Authors:** Abraham Degarege, Berhanu Erko, David M. Brett-Major, Bruno Levecke, Abebe Animut, Yohannes Negash, M. Jana Broadhurst, Tzeyu Michaud, Christopher R. Bilder

**Affiliations:** 1Department of Epidemiology, College of Public Health, University of Nebraska Medical Center, Omaha, NE 68198, USA; david.brettmajor@unmc.edu; 2Aklilu Lemma Institute of Pathobiology, Addis Ababa University, Addis Ababa 16417, Ethiopia; berhanu.erko@aau.edu.et (B.E.); abebe.animut@aau.edu.et (A.A.); yohannes.negash@aau.edu.et (Y.N.); 3Department of Translational Physiology, Infectiology and Public Health, Ghent University, 9000 Gent, Belgium; bruno.levecke@ugent.be; 4Department of Pathology, Microbiology and Immunology, University of Nebraska Medical Center, Omaha, NE 68198, USA; jana.broadhurst@unmc.edu; 5Department of Health Promotion, College of Public Health University of Nebraska Medical Center, Omaha, NE 68198, USA; tzeyu.michaud@unmc.edu; 6Department of Statistics, University of Nebraska-Lincoln, Lincoln, NE 68588, USA

**Keywords:** pooled urine samples, *Schistosoma haematobium*, sensitivity, urine reagent test strips

## Abstract

This study evaluated the performance of urine reagent strips (URSs) in detecting *Schistosoma haematobium* infection in individual and pooled urine samples. Between June 2022 and April 2023, 2634 urine samples (10 mL each) from school-age children (5–15 years) in 15 villages across Ethiopia’s Afar, Benishangul-Gumuz, and Gambella regions were tested using urine filtration microscopy (UFM) and URSs for blood, a marker of *S. haematobium* eggs. Pooled samples from 5, 10, 20, and 40 individuals (one positive, others negative) were examined with both methods. UFM results were used to calculate URSs’ sensitivity, specificity, and predictive values for detecting infection. A total of 2634 children were screened for *S. haematobium* infection. UFM detected *S. haematobium* eggs in 370 samples, while URSs identified infection in 414 children. URSs showed 64% sensitivity and 92% specificity for individual samples. The positive and negative predictive values for individual samples were 57% and 94%, respectively. Sensitivity for pooled samples ranged from 47% (pools of 40) to 53% (pools of 20). In pools with one positive sample, URSs misclassified 220 (50%), 109 (49.5%), 52 (47.0%), and 28 (50.9%) pools as negative for *S. haematobium* eggs for pool sizes 5, 10, 20, and 40, respectively. Sensitivity for individual samples was higher in children with heavy infection (92.5%) compared to light infection (55.9%), and sensitivity in pooled samples increased with infection intensity (*p* < 0.001). In conclusion, URSs may misclassify *S. haematobium* infection in children when samples are examined individually or in pools, potentially leading to unnecessary treatment or missed cases. However, URSs shows promise as a screening tool for detecting *S. haematobium* infection in areas with high infection intensity.

## 1. Introduction

*Schistosoma haematobium* affects over 112 million people worldwide, primarily in sub-Saharan Africa and the Middle East, where it is a major cause of urogenital schistosomiasis [1,2,3,4]. Although this disease is highly prevalent and poses severe health consequences and public health challenges in these resource-limited settings [1,2,3,4], diagnostic and treatment options are often insufficient. There remains a critical need for cost-effective diagnostic strategies to improve case detection and accurately estimate the prevalence of the disease at the population level, which would help in planning and monitoring mass drug administration programs.

One promising approach to address these challenges is group testing, also known as pooled testing, a diagnostic procedure designed to optimize resource use and reduce the time required for testing [5,6]. By combining samples from multiple individuals and testing them as a single unit, pooled testing significantly decreases the number of tests needed [5,6], making it an attractive option for large-scale screening of urogenital schistosomiasis. This approach has been successfully applied to various infectious diseases such as HIV, SARS-CoV-2, and Zika virus [7,8,9,10] and shows potential for large-scale diagnostic efforts for urogenital schistosomiasis [11,12], enabling more efficient detection and monitoring of the prevalence of the diseases in endemic regions. This innovative method could be particularly useful in resource-limited settings where individual testing may not be feasible due to cost or logistical constraints. In a small-scale study, we demonstrated that pooled testing based on the urine filtration microcopy (UFM) approach holds significant promise for estimating both the prevalence and intensity of infections at the population level [11]. Building on this work, we recently provided evidence that pooled testing with UFM can also be instrumental in rapidly identifying areas of high transmission, or “hot spots”, enabling targeted interventions and efficient allocation of resources [12].

The use of rapid diagnostic tools, such as urine reagent test strips (URSs) for blood detection in urine as a surrogate for the presence of *S. haematobium* eggs, presents opportunities to further enhance the efficiency of the pooled testing approaches for the diagnosis of *S. haematobium* infection [12,13]. *S. haematobium* infects the urinary system, causing hematuria (blood in the urine) due to bladder wall damage induced by the parasite’s eggs, leading to the presence of red blood cells or hemoglobin in the urine [4]. URSs can detect this condition, with hematuria severity (trace, mild, moderate, or severe) serving as an indicator of infection intensity [13]. However, hematuria may also result from menstruation, urinary tract infections, immune system or metabolic disorders, or other factors [14]. Therefore, confirmation through UFM or antigen-based tests is recommended for accurate diagnosis and assessment of infection intensity [14].

URSs offer quick turnaround times [3,4,14], which could significantly shorten the time needed for screening and identifying infections when integrated with pooled testing strategies. In addition, URSs do not require a microscope, electricity, or trained personnel, and are cheap and easy to transport and use in remote areas [14,15]. Moreover, the performance of URSs is less affected by the production and release of *S. haematobium* eggs [14,16]. Despite these characteristics that make URSs well suited for field screening of *S. haematobium* infection, studies have shown inconsistent results regarding the sensitivity (i.e., ranging from 16% to 100%) of the test in detecting infection with the parasite from individual samples [14,17]. The sensitivity of the test could be further affected in pooled samples due to the dilution effect on hematuria levels. However, there is a lack of robust evidence regarding the sensitivity and specificity of URSs for detecting *S. haematobium* infections in pooled urine samples, a critical knowledge gap relevant to optimizing large-scale surveys in endemic regions.

Recently, we reported results comparing the performance of UFM and URSs in identifying *S. haematobium* infection using 10 mL urine samples from individual and pooled samples [12]. However, that study employed a cascaded pooling approach (pools of 5 were used to make pools of 10, after which these pools of 10 were used to make pools 20 and pools of 20 to make pools of 40). Additionally, the study participants were recruited from only two regions in Ethiopia, and the sample size was insufficient to perform region-specific analyses. This is because the prevalence and intensity of *S. haematobium* infection can vary by region, potentially influencing the sensitivity of diagnostic tools used for detecting infection [12,13,16]. In the present study, we addressed these limitations by assessing the performance of URSs in detecting *S. haematobium* infection using a relatively large number of samples collected from three regions of Ethiopia with varying endemicity of urogenital schistosomiasis and pooled independently into groups of 5, 10, 20, and 40.

## 2. Materials and Methods

### 2.1. Study Population and Data Collection

Between June 2022 and April 2023, urine samples of at least 80 mL were collected from 2634 school-age children (5 to 15 years of age) living in selected villages of three regions of Ethiopia (Afar, Benishangul-Gumuz and Gambella) where there is a transmission of *S. haematobium* regardless of the occurrence of infection-related symptoms [11,12,18]. Literature information about the level of transmission of *S. haematobium* infection and recommendations from local administrative officials and guides, who considered accessibility and local control program activities in their efforts, were used to select the villages [11,12,13,18]. Urine samples were collected in nearby health posts, schools, or open fields of selected villages between 10:00 a.m. and 3:00 p.m. School-age children who assented and for whom parental consent was obtained were enrolled for the collection of 80 mL urine, 10 mL of which was first checked for the presence of blood as a proxy for *S. haematobium* infection using URSs (Human GmbH-Max-Planck-Ring 21, Wiesbaden, Germany) and then filtered through a polycarbonate filter membrane and examined under a microscope in the field for *S. haematobium* eggs. The remaining 70 mL urine sample was transferred to a vial containing 0.7 mL formalin (37% formaldehyde) and transported to the Medical Parasitology Laboratory of Aklilu Lemma Institute of Pathobiology of Addis Ababa University.

### 2.2. Urine Sample Pooling and Examination Procedure

Once the samples arrived at the laboratory, 10 mL was further checked for the presence of blood using a URS and filtered through a polycarbonate filter membrane and examined for *S. haematobium* eggs using a microscope following the same procedure as in the field. The remaining 60 mL of urine was strategically pooled and examined using URSs and UFM techniques. To this end, first each individual urine sample was declared either positive (eggs were found at least once) or negative for *S. haematobium* eggs (no eggs were found on both occasions) based on the obtained UFM results both in the field and at the laboratory. Then, plastic vials containing individual urine samples were arranged in rows of 5, 10, 20, and 40 samples. In each row, one positive sample was included, while the remaining samples were negative. Those samples with greater than zero *S. haematobium* eggs were randomly selected. Following that, in each row urine samples of 28 mL, 14 mL, 7 mL, and 3.5 mL from each individual sample were transferred to a 140 mL size vial to make pools of 5, 10, 20, and 40 samples, respectively. Finally, an aliquot of 10 mL of each pooled urine sample was tested for the presence of blood (using URSs) and *S. haematobium* eggs (using UFM).

### 2.3. Statistical Data Analysis

First, using the individual test results based on UFM in the field and lab as a reference (samples that tested positive in the field and/or lab were declared positive), we estimated the sensitivity, specificity and predictive values of the URSs. along with 95% Wald Confidence Interval (CI), in detecting *S. haematobium* infection in the individual urine samples across the age and gender groups, the regions where the children live, and the class of intensity of infection. The class of intensity of infection was grouped as light (UEC ≤ 50 eggs per 10 mL) and heavy (UEC > 50 eggs per 10 mL) based on WHO cut-off point [19]. Then, we compared the sensitivity of the URSs in detecting *S. haematobium* infection in pooled urine samples (i.e., the proportion of pooled samples that tested positive for the presence of blood) across all pool sizes (5, 10, 20, and 40), level of intensity of infection (level 1: mean urine egg count (UEC) of the individual samples making the pools per 10 mL urine ≤ 25th percentile; level 2: 25th percentile < mean UEC ≤ 50th percentile; level 3: 50th percentile < mean UEC ≤ 75th percentile; level 4: mean UEC > 75th percentile), and the regions where the samples were collected (Afar, Benishangul Gumuz, and Gambela) using a chi-square test for trend analysis [20]. An overall percent agreement was used to measure the percentage of total subjects where the UFM and the URSs results agree.

### 2.4. Ethical Consideration

This study was approved by the institutional review boards of the University Ne-braska Medical Center (IRB #0875-21-EP, 23 December 2021) and Aklilu Lemma Insti-tute of Pathobiology (Ref. No. ALIB IRB/63/2014/2021, 21 December 2021). The district health office, school authorities, and teachers were briefed on the study’s purpose and procedures. Written informed consent was obtained from the parents or guardians of children who agreed to participate. Children diagnosed with S. haematobium infection, based on UFM using 10 mL of urine, were treated with praziquantel (40 mg/kg body weight) free of charge.

## 3. Results

### 3.1. Performance of URSs in Detecting Infection in the Individual Urine Samples

A total of 2634 children (age 5–15, 41.7% girls) from 15 villages in three regions were screened for *S. haematobium* infection. *S. haematobium* eggs were detected in 14.1% of children using UFM, with most (83.5%) infections being light (≤50 eggs per 10 mL of urine). Prevalence was higher in older children (5–10 years: 12.2% vs. 11–15 years: 16.2%), boys (15.0% vs. 12.7%), and in Gambella (24.4% vs. Benishangul-Gumuz: 12.4% vs. Afar: 11.6%). Infections occurred in all villages, ranging from 0.5% to 38.0%.

When UFM was used for testing, *S. haematobium* eggs were detected in 370 samples, while the infection was identified in 414 children based on URSs. Using the results based on the UFM as a reference (Table 1), URSs showed a 64% percent positive agreement (“sensitivity”) in detecting children with *S. haematobium* infection and a 92% percent negative agreement (“specificity”) in declaring children not infected with the parasite when samples were examined individually. The overall percent agreement between UFM and URSs in detecting *S. haematobium* eggs was 88.15%. The positive and negative predictive values (PPV and NPV) of URSs in predicting children with and without *S. haematobium* infection were 57% and 94%, respectively. The sensitivity of the test was observed to be greater in males than in females; in children aged 12 and 13 compared to those aged 14, 15, or younger than 12 years old; for heavy compared to light intensity of infection; and in those who live in Afar than in Benishangul Gumuz or Gambella regions (Table 2). The specificity and NPVs of the test were comparable when deployed for detecting infection in males vs. females and in children of different age groups (Table 2).

### 3.2. Performance of URSs in Detecting Infection in Pooled Urine Samples

The sensitivity of URSs is summarized in Table 3. Values were comparable across the different pool sizes ranging from 47% (pool size 40) to 53% (pool size 20). The sensitivity of URSs in detecting pooled urine samples with *S. haematobium* eggs was 54.2% among samples collected in Afar, 47.1% in Benishangul Gumuz, and 48.3% in Gambella. However, the sensitivity significantly increased with an increase in the intensity level of infection (*p* < 0.001).

## 4. Discussion

This study investigated the utility of URSs as a diagnostic tool for detecting *S. haematobium* infection in individual and pooled urine samples. The prevalence of *S. haematobium* infection among the study participants was 14.1%, ranging from 0.5% to 38% across different villages. This is lower than the rates reported in previous studies conducted in the region [12,21]. The overall performance of the test in detecting infection from individuals (sensitivity = 64% and PPV = 57%) and pooled samples (sensitivity ≤ 53%) was not satisfactory. When samples were examined individually, URSs misclassified 134 children (out of 370), who tested positive for *S. haematobium* eggs using UFM, as uninfected and 178 uninfected children (out of 2264), who tested negative for *S. haematobium* eggs using UFM, as infected. Similarly, when samples from 2634 children were examined in pools of 5 (n = 440), 10 (n = 220), 20 (n = 110), and 40 (n = 55), with each pool containing one sample that tested positive for *S. haematobium* eggs based on UFM, URSs misclassified 220, 109, 52, and 28 pools as negative for *S. haematobium* eggs, respectively. Consequently, 220, 109, 52, and 28 infected children, depending on the pool size, may not have received treatment based on the URSs results. These results suggest that URSs may generally fail to accurately determine the status of infection when deployed for testing urine samples from children individually or in pools. Thus, a significant number of children could have received unnecessary treatment or miss relevant treatment if URSs are used to determine *S. haematobium* infection status. A number of studies also reported lower sensitivity of URSs in detecting *S. haematobium* infection when deployed for testing urine samples following individual or pooled testing procedures [3,12,13].

However, the sensitivity of URSs in detecting *S. haematobium* infection in individual samples was observed to be greater when used to diagnose these infections in individual samples collected from male children (67.7% vs. 59% in females) who were younger than 14 years (66.5% to 71.4% vs. 52.3% to 54.0% in those ≥14 years old), those who live in the Afar region (76.7% vs. 58.0% in Benishangul Gumuz or 55.4% in Gambella), and those with heavy-intensity (UEC > 50 eggs/10 mL urine) infection (92.5% vs. 55.9% in light-intensity infection). The sensitivity of URSs in detecting infection in pooled samples also increased with an increase in the intensity of infection, aligning with previous research findings [11,12,16,22,23,24]. These results suggest that while URSs may generally misclassify the presence of *S. haematobium* eggs in individual and pooled samples, they might be employed for rapidly exploring the presence of *S. haematobium* eggs in regions with high infection intensity and moderate endemicity levels.

While these results align with expectations, they have important practical implications. The observed increase in sensitivity suggests that pooled testing using URSs may help to detect and estimate the prevalence of *S. haematobium* infection at a population level with better accuracy in settings with higher infection intensities, especially when the pool size is small. Conversely, in areas with lower infection intensity/endemicity, pooled testing may have a low diagnostic accuracy even after adjustments to the pooling strategy. Similarly, while the variation in the sensitivity of URSs by region was anticipated [16], they impact interpretation of screening results across regions. Differences in URSs’ sensitivity across regions may reflect variations in infection intensity, urine composition, or environmental factors influencing diagnostic performance. These findings underscore that considering regional characteristics and endemicity contexts when deploying URSs for detecting *S. haematobium* infection could enhance the efficiency and accuracy of surveillance programs. Ultimately, understanding and addressing these regional variations can help optimize resource allocation and improve the effectiveness of schistosomiasis control efforts in diverse epidemiological settings.

The present study demonstrated that URSs may not be used to detect *S. haematobium* infection in individual or pooled urine samples with a good accuracy applicable to population risk assessment. However, only 10 mL of individual and pooled urine was examined in this study. Consequently, the potential influence of urine volume on the sensitivity of URSs for detecting infections could not be assessed. Furthermore, due to the limited number of samples collected in some individual villages, analysis at the village level was not conducted. This is a notable limitation, as the intensity and prevalence of infection, which are key factors influencing test performance, vary significantly due to local exposures. Such variability could provide further insights into the test’s applicability in diverse settings. Moreover, female children aged 14 to 15 years, who might have been post-menarche at the time of urine sample collection, could have influenced the results. The differential diagnosis of hematuria in children could also be related to kidney injury or failure due to poststreptococcal glomerulonephritis [25]. Future research is needed to address these limitations and expand the understanding of URSs application in large-scale epidemiological surveys. Specifically, studies should investigate the impact of urine volume on URSs’ sensitivity, assess its performance across varying levels of endemicity, and evaluate its utility during different program phases. These findings would help refine the use of URSs as a diagnostic tool for detecting *S. haematobium* infection in broader epidemiological and programmatic contexts for school-age children.

## 5. Conclusions

In conclusion, URSs may lead to missing a significant portion of children infected with *S. haematobium* and misdiagnose children who are not infected with the parasite as infected when urine samples are tested individually or in pools. However, URSs show potential as a screening tool for detecting *S. haematobium* infection in regions with high infection intensity and moderate endemicity levels. Further research is needed to optimize the application of URSs for diagnosing *S. haematobium* infection and for leveraging them as a rapid and cost-effective screening method in large-scale epidemiological surveys, enabling better surveillance and control of infections with the parasite in endemic areas.

## Figures and Tables

**Table 1 microorganisms-13-00510-t001:** Performance of URSs in detecting *S. haematobium* infection from individual samples.

		**UFM Results**		Total
		Positive	Negative	
**URSs Results**	Positive	236	178	414
	Negative	134	2086	2220
Total		370	2264	2634

UFM: urine filtration microscopy used as a reference; URSs: urine reagent strips.

**Table 2 microorganisms-13-00510-t002:** Performance of URSs in detecting *S. haematobium* infection from individual samples based on demographic status and class of intensity of infection alongside 95% Wald CI.

	Categories (n)	Sensitivity (95% CI)	Specificty (95% CI)	PPV (95% CI)	NPV (95% CI)
	≤10 (1399)	66.5 (59.4, 73.6)	92.9 (91.6, 94.4)	66.5 (60.0, 73.0)	92.9 91.4, 94.4)
	11 (150)	66.7 (44.9, 88.4)	93.9 (89.9, 98.1)	60.0 (38.5, 81.5)	95.4 (91.8, 99.0)
Age in years	12 (245)	71.1 (57.9, 84.3)	92.5 (89.5, 96.5)	68.1 (54.7, 81.3)	93.4 (90.0, 96.9)
	13 (171)	71.4 (56.4, 86.4)	91.2 (86.2, 95.8)	67.6 (53.0, 83.0)	92.5 (88.0, 96.9)
	14 (323)	52.3 (40.2, 64.4)	86.0 (81.8, 90.2)	48.6 (36.9, 60.3)	87.8 (83.8, 91.8)
	15 (346)	54.0 (37.9, 70.0)	94.0 (91.3, 96.6)	51.3 (35.3, 0.66.7)	94.5 (91.9, 97.0)
Gender	Females (1098)	59.0 (50.8, 67.2)	93.0 (91.4, 94.6)	55.4 (47.4, 63.4)	94.0 (92.5, 95.5)
	Males (1536)	67.0 (60.9, 73.1)	91.4 (89.9, 92.9)	58.0 (52.1, 63.9)	94.0 (92.7, 95.3)
	Afar (1143)	76.7 (69.5, 83.9)	88.3 (86.0, 90.0)	46.4 (39.8, 52.9)	96.7 (95.9, 98.1)
Region	Benishngul Gumuz (1052)	58.0 (48.6, 67.4)	91.6 (88.6, 94.6)	68.9 (60.3, 78.1)	87.1 (83.6, 90.6)
	Gambella (439)	55.4 (46.4, 63.6)	96.5 (95.3, 97.7)	69.2 (59.3, 78.5)	93.9 (92.4, 95.4)
Class of intensity of infection	Light (UEC ≤ 50 eggs per 10 mL) (290)	55.9 (50.2, 61.6)	Na	na	na
	Heavy (UEC > 50 eggs per 10 mL) (80)	92.5 (86.7, 98.3)	Na	na	na

Results from urine filtration microscopy used as a reference. PPV: positive predictive value; NPV: negative predictive value; na = not applicable.

**Table 3 microorganisms-13-00510-t003:** The diagnostic sensitivity of URSs in detecting positive pooled samples by pool size, the intensity level of the individual samples making the pools, and the region where the samples were collected.

Variables	Categories	Number of Pools with One Individual Sample Making Them Contain *S. haematobium* eggs	Number of Pools with *S. haematobium* eggs Based on URS (%)
Pool size	5	440	220 (50.0)
	10	220	111 (50.5)
	20	110	58 (53.0)
	40	55	27 (49.1)
*p*-value			0.834
Intensity level	1	249	71 (28.5)
	2	168	74 (44.0)
	3	202	111 (55.0)
	4	206	160 (78.0)
*p*-value			<0.001
Region	Afar	360	195 (54.2)
	Gambella	180	87 (48.3)
	Benishangul Gumuz	285	134 (47.1)
*p*-value			0.067

## Data Availability

The original contributions presented in this study are included in the article/Appendix A. Further inquiries can be directed to the corresponding author.

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
