# Peer review of "Performance of Urine Reagent Test Strips in Detecting Schistosoma haematobium Infection in Individual and Pooled Urine Samples"

_microorganisms, 2025, doi:10.3390/microorganisms13030510_

Round 1

Reviewer 1 Report

Comments and Suggestions for Authors

The study presents a novel method for diagnosing Schistosoma haematobium infection using pooled urine samples in endemic areas of Ethiopia. This approach appears to introduce an innovative technique for the rapid detection of S. haematobium in African countries. Below are some comments and suggestions for improvement:

 1. The authors mention the "urine reagent test strips" method in the manuscript title but fail to provide a detailed introduction to the reagent strips in the text. For instance, is this an ELISA-based detection method? What is the specific target of the detection? How is the infection level of the samples confirmed? Additionally, what are the standard criteria used for detection? These details are crucial for understanding the methodology and its validity.

 2. As shown in Table 1, 414 samples tested positive using the urine reagent strip (URS) method, while 370 tested positive using urine filtration microscopy (UFM) out of 2,634 samples. However, the authors should first clarify the infection status of all samples before drawing conclusions about sensitivity and specificity. It is recommended to retrospectively review the data to confirm whether the samples were infected or had undergone prior treatment. This step is essential for ensuring the accuracy and reliability of the results.

 3. Some minor errors need to be addressed to meet the publication standards. For example, the scientific name ‘Schistosoma haematobium’ should be italicized consistently throughout the manuscript. Additionally, there are missing punctuation marks, such as in the last sentence of the discussion section. These small but important details should be corrected to enhance the overall quality of the paper.

Author Response

Dear Reviewer 

We would like to thank you for your careful review and constructive comments, which have helped to improve the manuscript. We have made changes to the manuscript based on your suggestions and describe these changes in the below paragraphs. We hope that you will find our responses acceptable, and we look forward to your decision.

Reviewer 1

The study presents a novel method for diagnosing Schistosoma haematobium infection using pooled urine samples in endemic areas of Ethiopia. This approach appears to introduce an innovative technique for the rapid detection of S. haematobium in African countries. Below are some comments and suggestions for improvement:

  1. The authors mention the "urine reagent test strips" method in the manuscript title but fail to provide a detailed introduction to the reagent strips in the text. For instance, is this an ELISA-based detection method? What is the specific target of the detection? How is the infection level of the samples confirmed? Additionally, what are the standard criteria used for detection? These details are crucial for understanding the methodology and its validity.
  • Thank you, we have added a paragraph describing the targets of urine reagent test strips and the strategy to determine infection intensity using the test as suggested.
  • The added text in the introduction (page 2) reads: “ haematobium infects the urinary system, causing hematuria (blood in the urine) due to bladder wall damage induced by the parasite's eggs, leading to the presence of red blood cells or hemoglobin in the urine [4]. URS can detect this condition, with hematuria severity (trace, mild, moderate, or severe) serving as an indicator of infection intensity [13]. However, hematuria may also result from menstruation, urinary tract infections, immune system or metabolic disorders or other factors [14]. Therefore, confirmation through UFM or antigen-based tests is recommended for accurate diagnosis and assessment of infection intensity [14].
  1. As shown in Table 1, 414 samples tested positive using the urine reagent strip (URS) method, while 370 tested positive using urine filtration microscopy (UFM) out of 2,634 samples. However, the authors should first clarify the infection status of all samples before drawing conclusions about sensitivity and specificity. It is recommended to retrospectively review the data to confirm whether the samples were infected or had undergone prior treatment. This step is essential for ensuring the accuracy and reliability of the results.
  • Thank you, we have added a paragraph describing the infection status of the samples as suggested.
  • The added text in the results (page 4) reads: “A total of 2,634 children (ages 5–15, 41.7% girls) from 15 villages across three regions were screened for haematobium infection. S. haematobium eggs were detected in 14.1% of children using UFM, with most (83.5%) infections being light (≤ 50 eggs per 10 mL of urine). Prevalence was higher in older children (5–10 years: 12.2% vs. 11–15 years: 16.2%), boys (15.0% vs. 12.7%), and in Gambella (24.4% vs. Benishangul-Gumuz: 12.4% vs. Afar: 11.6%). Infections occurred in all villages, ranging from 0.5% to 38.0%.”
  • Regarding data review to confirm infection, although UFM is the current standard technique for diagnosing haematobium, it’s not 100 accurate in detecting infection. So, we have also referred sensitivity as percent positive agreement and specificity as percent negative agreement. And also added overall agreement results between the two tests in detecting infection.

The revised results (added results) read as “Using the results based on the UFM as a reference (Table 1), URS showed a 64% percent positive agreement (“sensitivity”) in detecting children with S. haematobium infection and a 92% percent negative agreement (“specificity”) in declaring children not infected with the parasite when samples were examined individually. The overall percent agreement between UFM and URS in detecting S. haematobium eggs was 88.15%.”

  1. Some minor errors need to be addressed to meet the publication standards. For example, the scientific name ‘Schistosoma haematobium’ should be italicized consistently throughout the manuscript. Additionally, there are missing punctuation marks, such as in the last sentence of the discussion section. These small but important details should be corrected to enhance the overall quality of the paper.
  • Thank you, we have edited the manuscript for typos in grammar and article usage, italicization of scientific names and punctuation.

Reviewer 2 Report

Comments and Suggestions for Authors

The manuscript is carefully written and focuses on the contemporary problem of diagnosing Schistosoma haematobium in endemic and resource-limited countries. The results obtained from the conducted studies indicate a low sensitivity of URS in diagnosing Schistosoma haematobium, especially in cases of low invasion and in pooled samples. The authors rightly draw attention to the consequences resulting from the use of the described URS method, but also appreciate its potential. I have no significant criticisms, but I will make a few suggestions that could increase the value of the manuscript.
1. Title. Schistosoma haematobium should be written in italics.
2. Abstract. In the first line of the abstract, it would be worth expanding the abbreviation ‘URS’, which appears for the first time in the text.
3. Keywords. It is better to arrange the keywords in alphabetical order.
4. Introduction. In the second paragraph of the introduction, in the sentence concerning the use of the sample pooling method in other infectious diseases, it is worth mentioning a few specific ones.
5. Materials and methods. Study population and data collection. Were there any other criteria for selecting the study group apart from age and area of residence (e.g. occurrence of symptoms)?
6. Materials and methods. Study population and data collection. The number of people studied was given in the results, it could also be given in this section.
7. Materials and methods. Study population and data collection. There is no information about the manufacturer of the URS strips used in this study.
8. Materials and methods. Urine sample pooling and examination procedure. There was one positive sample in each row. Was this positive sample selected randomly in the context of the number of Schistosoma haematobium eggs detected in it?
9. Discussion. There is no sentence indicating whether the percentage of positive results for S. haematobium in the study group using the reference method (UFM) is consistent with scientific reports/information on the prevalence of S. haematobium in the study region?

Comments on the Quality of English Language

 The English could be improved to more clearly express the research

Author Response

Dear Reviewer 

We would like to thank you for your careful review and constructive comments, which have helped to improve the manuscript. We have made changes to the manuscript based on your suggestions and describe these changes in the below paragraphs. We hope that you will find our responses acceptable, and we look forward to your decision.

The manuscript is carefully written and focuses on the contemporary problem of diagnosing Schistosoma haematobium in endemic and resource-limited countries. The results obtained from the conducted studies indicate a low sensitivity of URS in diagnosing Schistosoma haematobium, especially in cases of low invasion and in pooled samples. The authors rightly draw attention to the consequences resulting from the use of the described URS method, but also appreciate its potential. I have no significant criticisms, but I will make a few suggestions that could increase the value of the manuscript.

  1. Schistosoma haematobium should be written in italics.
  • Thank you, we have italicized “Schistosoma haematobium” in the title.
  1. In the first line of the abstract, it would be worth expanding the abbreviation. ‘URS’, which appears for the first time in the text.
  • Thank you, we have added the full text for URS, “urine reagent strips” in the abstract.
  1. It is better to arrange the keywords in alphabetical order.
  • Thank you, we have arranged the keywords in alphabetical order.
  1. In the second paragraph of the introduction, in the sentence concerning the use of the sample pooling method in other infectious diseases, it is worth mentioning a few specific ones.
  • Thank you, we have listed infectious diseases to which the pooled testing strategy has been successfully applied. The revised text reads as, “This approach has been successfully applied to various infectious diseases such as HIV, SARS CoV-2, Zika virus” (Introduction, page# 2, paragraph # 2).
  1. Materials and methods. Study population and data collection. Were there any other criteria for selecting the study group apart from age and area of residence (e.g. occurrence of symptoms)?
  • Age (5 to 15 years) and area of residence were the only inclusion criteria used for this study. The revised text in the methods section reads as follows, “Between June 2022 and April 2023, urine samples of at least 80 mL were collected from 2,634 school-age children (5 to 15 years of age) living in selected villages of three regions of Ethiopia (Afar, Benishangul-Gumuz and Gambella) where there is transmission of S. haematobium regardless of the occurrence of infection related symptoms [11,12,18].”
  1. Materials and methods. Study population and data collection. The number of people studied was given in the results, it could also be given in this section.
  • Thank you, we have provided the number of people participated in this study in the methods section. “Between June 2022 and April 2023, urine samples of at least 80 mL were collected from 2,634 school-age children (5 to 15 years of age) living in selected villages of three regions of Ethiopia (Afar, Benishangul-Gumuz and Gambella) where there is transmission of S. haematobium regardless of the occurrence of infection related symptoms [11,12,18].”
  1. Materials and methods. Study population and data collection. There is no information about the manufacturer of the URS strips used in this study.
  • Thank you, we have provided the manufacturer information for the URS in the methods. It reads “School-aged children……was first checked for the presence of blood as a proxy for S. haematobium infection using URS (Human GmbH-Max-Planck-Ring 21, Wiesbaden, Germany) (Page #3; paragraph# 1).
  1. Materials and methods. Urine sample pooling and examination procedure. There was one positive sample in each row. Was this positive sample selected randomly in the context of the number of Schistosoma haematobium eggs detected in it?
  • Thank you. Selection of positive samples was done on a a random basis. We have added this information to the methods. It reads “Those samples with greater than zero haematobium eggs were randomly …..”(page # 3, paragraph # 2).
  1. There is no sentence indicating whether the percentage of positive results for S. haematobium in the study group using the reference method (UFM) is consistent with scientific reports/information on the prevalence of S. haematobium in the study region?
  • We have compared the prevalence of S. haematobium infection estimated in the current study with previous reports. The added text in the discussion reads as “ The prevalence of S. haematobium infection among the study participants was 14.1%, ranging from 0.5% to 38% across different villages. This is lower than the rates reported in previous studies conducted in the region [12, 21]”.